# Genetic Determinants of 25-Hydroxyvitamin D Concentrations and Their Relevance to Public Health

**DOI:** 10.3390/nu14204408

**Published:** 2022-10-20

**Authors:** Elina Hyppönen, Karani S. Vimaleswaran, Ang Zhou

**Affiliations:** 1Australian Centre for Precision Health, Clinical and Health Sciences, University of South Australia, Adelaide, SA 5001, Australia; 2South Australian Health and Medical Research Institute, Adelaide, SA 5001, Australia; 3Hugh Sinclair Unit of Human Nutrition, Department of Food and Nutritional Sciences, University of Reading, Reading RG6 6DZ, UK; 4The Institute for Food, Nutrition and Health (IFNH), University of Reading, Reading RG6 6DZ, UK

**Keywords:** 25-hydroxyvitamin D, vitamin D, genetic risk, heritability, personalized supplementation, genome-wide association study, Mendelian randomization

## Abstract

Twin studies suggest a considerable genetic contribution to the variability in 25-hydroxyvitamin D (25(OH)D) concentrations, reporting heritability estimates up to 80% in some studies. While genome-wide association studies (GWAS) suggest notably lower rates (13–16%), they have identified many independent variants that associate with serum 25(OH)D concentrations. These discoveries have provided some novel insight into the metabolic pathway, and in this review we outline findings from GWAS studies to date with a particular focus on 35 variants which have provided replicating evidence for an association with 25(OH)D across independent large-scale analyses. Some of the 25(OH)D associating variants are linked directly to the vitamin D metabolic pathway, while others may reflect differences in storage capacity, lipid metabolism, and pathways reflecting skin properties. By constructing a genetic score including these 25(OH)D associated variants we show that genetic differences in 25(OH)D concentrations persist across the seasons, and the odds of having low concentrations (<50 nmol/L) are about halved for individuals in the highest 20% of vitamin D genetic score compared to the lowest quintile, an impact which may have notable influences on retaining adequate levels. We also discuss recent studies on personalized approaches to vitamin D supplementation and show how Mendelian randomization studies can help inform public health strategies to reduce adverse health impacts of vitamin D deficiency.

## 1. Introduction

Interest in the genetic architecture of 25-hydroxyvitamin D (25(OH)D) has been active during the past couple of decades, promoted by heritability estimates from twin studies which suggest that up to 80% of the variability in 25(OH)D concentrations might be explained by genetic variation in some populations [1,2,3,4,5]. Also an evolutionary perspective provides strong cues about the importance of genetic variation for vitamin D metabolism [6], as differences in skin colour are believed to have evolved at least in part as an adaptation to ultraviolet B radiation exposure during migration to more northern latitudes, where reduction in skin pigmentation became critical to vitamin D synthesis. Important insights into the genetic architecture of 25(OH)D concentrations have been obtained from genome-wide association studies (GWASs) which have used information across thousands of genomes to find polymorphisms which are statistically associated with 25(OH)D. For this paper, we systematically looked through the GWAS literature for genes that influence serum 25(OH)D levels. We describe key variants for which evidence has been provided from several independent studies and address the public health importance and some of the uses of this information.

## 2. Materials and Methods

For the systematic search of GWASs, we searched the MEDLINE, Embase, Cochrane, CINAHL and NHGRI-EBI GWAS catalogue [7] databases for original studies and meta-analyses of studies performed in humans and published in English from inception to February, 2022. The search terms used were (“vitamin D” OR “calcidiol” OR “25-hydroxyvitamin D” OR “25(OH)D”) AND (“genome-wide association study” OR “genome-wide association scan” OR “genome-wide association analysis”) along with the expanded MeSH search terms (in titles, abstracts, or keywords). The search identified 791 publications in total. After excluding duplicate entries, 522 publications remained, of which 29 were relevant. These were further scrutinized to identify sample overlap. We also scrutinized references within the selected articles, and from studies otherwise known to the authors, with evidence on gene function queried using the gene ontology (GO) resource (http://geneontology.org/, accessed on 27 September 2022), KEGG (https://www.genome.jp/kegg/, accessed on 27 September 2022), ConsensusPathDB (http://cpdb.molgen.mpg.de/, accessed on 27 September 2022) and other NCBI (https://www.ncbi.nlm.nih.gov/guide/all/, accessed on 27 September 2022) databases.

## 3. Results

### 3.1. Genome-Wide Association Studies on 25(OH)D

Our literature search identified 29 published GWASs that looked for SNPs associated with 25(OH)D [8,9,10,11,12,13,14,15,16,17,18,19,20,21,22,23,24,25,26,27,28,29,30,31,32,33,34,35,36], although many of these analyses were conducted using overlapping samples. Most of the studies only include adult participants of white European ancestry (*n* = 6722 to 443,374) [8,10,14,16,19,20,22,24,26,29,30,32,33,34,36]. There were seven studies including data from transethnic analyses [15] and studies with small to modest sample sizes (*n* = 697 to 9823) which had been conducted using data from African Americans (*n* = 697 in the discovery sample) [25], African descent in the UK (*n* = 9354) [35], Hispanic (*n* = 1190) [9] and Asian populations (*n* = 1387 to 9823) [13,23,28,35]. There were also five small GWASs on children/toddlers/new-borns [11,12,18,21,27].

Three loci, including *DHCR7, CYP2R1*, and *GC* were consistently reported across European [8,10,14,16,19,20,22,24,26,34] and several non-European GWASs [11,13,15,18,21,23,25,27,28]. These loci were also confirmed in GWAS conducted in children/toddlers/new-borns [11,12,18,21,27]. These genes fit with existing evidence of the involvement of their corresponding proteins in the vitamin D metabolic pathway. The *DHCR7* gene encodes the 7-dehydrocholesterol reductase, which is an enzyme that converts dehydrocholesterol to cholesterol in the skin, and affects the substrate availability for vitamin D3 synthesis, which is a precursor of 25(OH)D [37] (Figure 1). The *CYP2R1* gene encodes the enzyme in the cytochrome P-450 family 2R1, and is the primary 25-hydroxylase in the liver, converting vitamin D to 25(OH)D. In the circulation, vitamin D metabolites including 25(OH)D are mainly found bound to vitamin D binding protein (encoded by *GC*), which in most of the GWASs to date, have come up with the strongest signal for 25(OH)D. GWAS on non-European cohorts have also reported some novel loci (e.g., *FOXA2/SSTR4* [13], *HSPG2* [25], *TINK* [25] and *KIF4B* [15]) which have not been identified in GWAS studies on white European ancestry. Independent replication is lacking with respect to most of these novel loci, and many have no clear link with the vitamin D metabolic pathway. One exception is *CYP2J2*, discovered in a multi-ethnic cohort of 942 pregnant women of Malay, Indian and Chinese ancestry [27]. *CYP2J2* encodes an enzyme (Cytochrome P-450 family 2, subfamily J, polypeptide 2) shown in vitro to act as a vitamin D hydroxylase [38]. There is thus a strong biological basis for this association. It is notable that variants coding the α1-hydroxylase (*CYP27B1*) or the vitamin D receptor (*VDR*) have not been identified by the GWASs on 25(OH)D conducted to date. Concentrations of active 1,25(OH)_2_D in the circulation are ~1000 times lower than those of 25(OH)D. Therefore, it is possible that differences arising from the conversion of 25(OH)D to 1,25(OH)_2_D, or those reflecting *VDR* related differences in the ‘usage’ of 1,25(OH)_2_D, may simply be too small to detect.

#### Variants with Evidence for Replicated Association with 25(OH)D

In the largest GWAS published to date, identified variants for 25(OH)D concentrations were enriched in genes in the metabolic vitamin D pathway, lipid and lipoprotein pathways, and pathways related to skin properties. The liver, the brain and the skin were identified as the top three locations where 25(OH)D-associated loci may exert their actions [20]. In Table 1 we present more detailed information for the 35 common autosomal SNPs which were identified as top hits in the GWAS conducted in the UK Biobank [20], and which were associated in a consistent direction with serum 25(OH)D concentrations in the SUNLIGHT consortium meta-analyses [16,20,39]. In addition, there were 70 common variants that could not be replicated in the SUNLIGHT consortium meta-analyses. These included variants that are known to be pleiotropic (i.e., affect multiple traits) and/or affect cholesterol metabolism (e.g., *PCSK9, LIPC, ABCA1, CETP, APOE, APOB, APOC1, LIPG* and *LDLR*). It is possible that some of the differences between the UK Biobank findings and the SUNLIGHT consortium meta-analysis relates to the body mass index adjustment by the SUNLIGHT, which reduces the likelihood of adipose tissue related variants (reflecting differences in storage capacity) being detected. It should be noted, however, that replication does not imply causality, and the possible connection which we have identified with vitamin D metabolism/function is only based on literature available to date and may not fully describe the connection with 25(OH)D concentrations.

For many of the variants which have a replicated association with 25(OH)D we found some evidence that was compatible with a role in the vitamin D pathway (Figure 1, Table 1). Several of the variants were related to lipid levels, while others had been linked with skin integrity, suggesting a possible link with substrate availability for conversion to the circulating 25(OH)D metabolite. Despite body mass index adjustment in the SUNLIGHT consortium meta-analyses, for several of the replicating variants we observed suggested links with adiposity or muscle mass, which may be because these tissues serve as storage sites for 25(OH)D. In addition to confirming the role of 24-hydroxylation in the inactivation and removal of vitamin D metabolites, GWAS identified variants in *UGT1A5* and *SULT2A1* as potentially relevant. This suggests that sulphonation and glucuronidation pathways, which conjugate sulphur and glucuronide with vitamin D metabolites, are relevant for 25(OH)D excretion and/or recycling.

### 3.2. Heritability and the Genetic Contribution to the Prevalence of Deficiency

25(OH)D is a commonly used indicator of ‘vitamin D status’ with much of the concentrations determined based on availability of sunlight induced skin synthesis, with contributions from supplement intake and diet. According to twin- and family-based studies, there is great variability in the heritability of 25(OH)D, with overall estimates ranging from 28% to 80% [1,2,3,4]. As ‘heritability’ merely reflects the proportion of total variance that can be explained by genetic factors, it will be the higher with lower environmental contributions (total variance = genetic variance + environmental variance). Indeed, the highest heritability rates are seen in populations measured during winter, when contributions from sunlight synthesis (and hence, the environmental effects) are at their lowest. This may explain why a study on 510 middle-aged male twins found heritability estimates to be ~70% when assessed in winter compared to negligible (~0%) during summer [84]. Also a twin study of Hispanics and African Americans, reported heritabilities of 23%, 28% and 41% for 25(OH)D levels from data taken from California, Texas, and Colorado (*n* = 1530 individuals from 130 families), respectively [85]. These differences were broadly reflective of geographical location, such that the higher the latitude (and the less sunlight exposure) the higher the estimated heritability. However, seasonal differences in heritability have not been consistently observed, and for example in the recent 25(OH)D GWAS [20], heritability was higher in summer than in winter (0.19 vs. 0.10, respectively). Indeed, another way to estimate heritability is to use information from unrelated individuals using GWAS data (i.e., SNP heritability) and in the large UK Biobank GWAS on white Europeans, SNP heritability for serum 25(OH)D concentrations was estimated to be 13–16% [19,20]. Of practical relevance is the extent to which these genetic variants affect the circulating 25(OH)D levels and the prevalence of vitamin D deficiency. In Table 2**,** we show the distribution of 25(OH)D concentrations based on data from the 35 replicating variants in the UK Biobank. The odds of low concentrations are about halved for individuals in the highest 20% of vitamin D genetic risk score compared to the lowest quintile. Again, these genetic associations appear to be slightly stronger during summer compared to winter, possibly reflecting genetic differences in vitamin D skin synthesis. The overall difference in the mean 25(OH)D between individuals in the highest vs. lowest quartile of the GRS is about 9 nmol/L [40], which is similar to the association seen with self-reported use of vitamin D supplementation in the UK Biobank during winter (9.7 nmol/L) [84]. This suggests that if the supply from sunlight or diet is limited, differences in 25(OH)D concentrations by higher genetic burden may be of clinical relevance.

### 3.3. Genetic Differences in Response to Supplementation and the Need for Personalized Approaches

There were 25 independent loci which were suggestive of gene–environment (GxE) interaction in the recent GWAS [20], suggesting that the size of the genetic association with 25(OH)D can vary by environmental factors influencing serum 25(OH)D concentrations. For five loci, including *CYP2R1* and *SEC23A* there was a genome-wide significant interaction with season [19,20], where the carriers of 25(OH)D-lowering alleles appeared to be less responsive to season compared to non-carriers. This could suggest that some individuals may be more prone to low serum 25(OH)D levels regardless of the season of measurement [19]. In an earlier genome-wide GxE analysis, carriers of 25(OH)D-lowering allele at the *CYP2R1* locus were less responsive to dietary vitamin D intake [16]. A similar interaction with vitamin D lowering alleles has also been observed in the context of the *GC* locus and vitamin D supplementation [87], of vitamin D3-fortified bread and milk consumption [86,87] and UVB treatment [88,89].

There has been recent interest in genetic risk scores (GRS) that combine variants according to their vitamin D lowering alleles, and which look into whether individuals with genetically low 25(OH)D are less or more responsive to treatments for correcting low vitamin D status [87,88,89]. One study used a GRS combining variants in the *CYP2R1* and *GC* loci, and reported a somewhat more modest (~23%) increase in serum 25(OH)D concentrations in response to UVB treatment for individuals carrying four risk alleles compared to the 54% increase for those carrying no risk alleles [88]. They also found that individuals with four risk alleles benefitted the least from the consumption of vitamin D3–fortified bread and milk during this 6-month study [88]. GRS for 25(OH)D has been suggested to be useful for guiding the screening and treatment for vitamin D deficiency. This was tested in a recent study [26], where participants with serum 25(OH)D < 50 nmol/L were recommended to take vitamin D supplements, adjusting the dosage according to their genetic risk. Again, this study used a simple GRS (two SNPs only, taken from *GC* and *CYP2R1)*, and the individuals with three or four 25(OH)D-lowering alleles were instructed to take 50 µg (2000IU) per day, those with one to two risk alleles to take 20–30 µg/day and those with no risk alleles, 10–20 µg per day. In their study, recommendation to take 50 µg (2000IU) per day over 4 months was enough to reduce the gap between individuals carrying three or four risk alleles and those with no risk alleles both with respect to serum 25(OH)D concentration and the prevalence of 25(OH)D < 50 nmol/L. However, the prevalence of 25(OH)D < 50 nmol/L remained elevated for those with two risk alleles compared to no risk alleles. While these results are very interesting and even promising, they are tentative, as the higher vitamin D intakes were achieved by recommendations, and not by testing in a placebo controlled, and randomized context. This was also a relatively small study (*n* = 10 to *n* = 36 per treatment group), so further trials with appropriate controls and a larger sample are required to examine possible benefits and effective approaches for personalized vitamin D supplementation. Given more profound genetic adaptations to differences in vitamin D intakes (‘vitamin D scarcity’) are possible, more research is also needed to establish target levels reflecting ‘optimal’ 25(OH)D concentrations and supplementation approaches for specific population groups, including indigenous Arctic and Tropical peoples [90].

### 3.4. Mendelian Randomization to Establish Evidence for Causal Effects of 25(OH)D

With the identification of genetic variants associated with serum 25(OH)D concentrations, it has become possible to use Mendelian randomization (MR) to examine evidence for the causal effect of vitamin D on other traits. This method is sometimes called the “natures controlled trial”, as assuming random allocation of genetic variants during the gamete formation, individuals are randomized on different exposure groups based on the genetic variants they carry. Reliable causal inference based on MR studies is conditional to some key method assumptions, and where these hold, this method can help avoid bias due to confounding and reverse causation which more strongly affect other types of observational studies [91]. MR analyses on 25(OH)D have used several strategies, and many of the studies have restricted the variants used to those in the actual vitamin D pathway (including *DHCR7*, *CYP2R1*, *GC* and *CYP24A1*). With additional loci being discovered for serum 25(OH)D [19,20], MR studies now commonly incorporate these new loci into the analyses. While the inclusion of additional loci can improve statistical power, it is also important to keep in mind the potential for pleiotropic effects that these variants could bring into the models and which could bias the MR analysis [92].

MR studies on 25(OH)D have been conducted across a wide range of outcomes, with evidence supportive of a causal effect seen for multiple sclerosis [93], type 2 diabetes [94] and hypertension [95]. However, many of the newly discovered variants do not have clear or known function with respect to vitamin D metabolism, and some appear pleiotropic, with associations with other traits, such as BMI, and lipid measures. One approach to alleviate concerns relating to pleiotropy and residual genetic confounding affecting variant selection, is to restrict the analyses to variants which have consistent replicating association with 25(OH)D concentrations [39], as would be the case if we use the 35 SNPs described above. However, even there, pleiotropy is likely to remain a concern, and sensitivity analyses using different sets of variants and different analytical approaches will be required to help to assess the robustness of the findings. A multivariable MR approach, which directly accounts for pleiotropic effects by modelling the genetic effects on 25(OH)D simultaneously with pleiotropy related indicators, may also be helpful. However, to allow for the use of this approach, the relevant pleiotropic pathways will need to be hypothesized and relevant information must be available for the analyses. In the context of threshold effects, rigorously conducted MR studies that take into account non-linearity can increase the value of the genetic approach for vitamin D research, as evidence for an effect may only be seen at very low or high levels [96,97]. Recruiting people with vitamin D deficiency to supplementation trials is an important challenge, and often studies test the effects of supplementation in individuals who already have adequate concentrations, and who are typically allowed to take over-the-counter supplements [96,97]. Evidence for benefits with rectifying vitamin D deficiency with respect to outcomes such as mortality [98,99], cardiovascular disease [39] and dementia [100] has been obtained from recent studies using non-linear or stratified MR approaches. For dementia, evidence for a causal effect of vitamin D had already been provided by linear MR studies [101,102,103], while effects on mortality had been supported by RCT meta-analyses [104], but the non-linear studies in both contexts suggest that the benefits of increasing levels may be largely confined to the correction of clinical deficiency. These findings provide important insight into strategies that are likely to provide the greatest benefits, suggesting that large-dose supplementation is unlikely to be required, but population level strategies such as food fortification, which can ensure at least minimal intakes and eradicate severe deficiency across the range of population groups, is likely to work.

## 4. Conclusions

Vitamin D status is in part determined by genetic variation and GWAS studies have identified a large number of variants that are associated with circulating 25(OH)D concentrations. Some of them are linked to the actual vitamin D metabolic pathway, and others to lipid metabolism and skin properties. In terms of methodology, they may provide MR studies with the means to measure the various determinants of serum 25(OH)D concentrations. Further research is needed to understand how such genetic information may be used to personalize vitamin D supplementation and prevent vitamin D deficiency.

## Figures and Tables

**Figure 1 nutrients-14-04408-f001:**
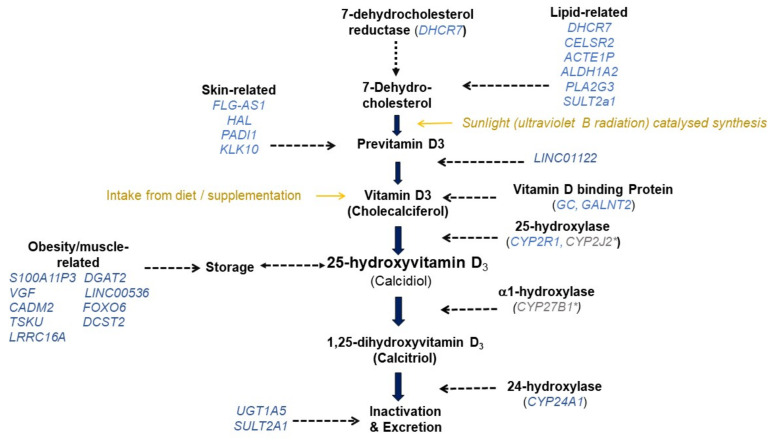
Possible role of selected replicated variants in the vitamin D pathway. * Indicates candidate genes with a confirmed role in vitamin D metabolism but which were not among the replicating variants. For the full list of single nucleotide polymorphism relating to each gene, please refer to Table 1.

**Table 1 nutrients-14-04408-t001:** Genetic variants with replicating evidence for an association with 25-hydroxyvitamin D concentrations.

Gene (SNP)	CHR	
*PEX10* (rs6671730)	1	*PEX10* encodes a protein involved in import of peroxisomal matrix proteins. Mutations in *PEX10* gene have led to Zellweger syndrome [40] and osteopenia [41], for which vitamin D supplementation has been the treatment.
*PADI1* (rs35408430)	1	*PADI1* encodes an enzyme, which catalyses the post-translational deimination of proteins by converting arginine residues into citrullines in the presence of calcium ions [42]. Deimination by PADIs occurs during epidermal differentiation [43], with possible influence on skin properties [20].
*FOXO6* (rs7522116)	1	*FOXO6* encodes a protein that has been predicted to enable DNA-binding transcription factor activity, and RNA polymerase related DNA binding activity [44]. FoxO6 expression is downregulated in the brain of dietary obese mice [45].
*CELSR2* (rs7528419)	1	*CELSR2* encodes the cadherin EGF LAG seven-pass G-type receptor 2 that is involved in contact-mediated communication, with cadherin domains acting as homophilic binding regions and the EGF-like domains involved in cell adhesion and receptor-ligand interactions [46].
*FLG-AS1* (rs1933064)	1	FLG antisense RNA 1 (*FLG-AS1*) is an RNA Gene that is affiliated with the long non-coding RNA class. Skin pigmentation-related diseases such as Ichthyosis Vulgaris [47] and Peeling Skin Syndrome 6 [48] have been shown to be associated with FLG-AS1.
*DCST2* (rs76798800)	1	*DCST2* gene encodes the DC-STAMP domain containing 2 protein that has been shown to be an important regulator of osteoclast cell-fusion in bone homeostasis [49]. *DCST2* gene is associated with early length and adult height [50].
*GALNT2* (rs6672758)	1	*GALNT2* gene encodes the polypeptide N-acetylgalactosaminyltransferase 2 which is a member of the glycosyltransferase 2 protein family and which has been linked to post-translational modification of vitamin D-binding protein [51].
*LINC01122* (rs727857)	2	*LINC01122* gene is an RNA gene that is affiliated with the lncRNA class [52]. *LINC01122* was one of the 989 differentially expressed genes which was significantly enriched in vitamin D3 biosynthesis [53].
*CPS1* (rs1047891)	2	*CPS1* gene encodes the carbamoyl-phosphate synthase 1 which is a mitochondrial enzyme that catalyses synthesis of carbamoyl phosphate from ammonia and bicarbonate [54].
*UGT1A5* * (rs2012736)	2	*UGT1A5* gene encodes the UDP glucuronosyltransferase family 1 member A5 which has been shown to transform small lipophilic molecules, into water-soluble, excretable metabolites [55]. Related isoenzymes have been identified as catalysts for 25(OH)D3 glucuronidation in the human liver [56].
*CADM2* (rs6782190)	3	*CADM2* gene encodes the cell adhesion molecule 2 which is a member of the synaptic cell adhesion molecule 1 family [57]. In animal studies, *CADM2* is associated with metabolic traits [58], suggesting possible influence on vitamin D concentrations through its effect on obesity and storage capacity of 25(OH)D.
*GC* (rs705117, rs1352846)	4	*GC* gene encodes the vitamin D binding protein which binds to vitamin D and its plasma metabolites and transports them to target tissues [59].
*CARMIL1/LRRC16A* (rs78151190)	6	*CARMIL1* gene encodes the capping protein regulator and myosin 1 linker 1 with a role in actin filament network formation [60]. Approximately 10% of muscle tissue consists of actin, providing a possible link with 25(OH)D through storage capacity.
*VGF* (rs75741381)	7	*VGF* gene encodes a protein that is expressed in neuroendocrine cells and is upregulated by nerve growth factor [61]. *VGF* has been linked with appetite control [62], and diet-induced obesity [63], with a possible link through storage capacity.
*LINC00536* (rs12056768)	8	*LINC00536* gene interacts with Wnt3a/β-Catenin signalling [64]. Wnt/β -Catenin signalling is an important signalling pathway in regulating adipose tissue lipogenesis with a possible link with 25(O)D through storage capacity.
*GRID1* (rs77532868)	10	*GRID1* gene encodes the glutamate ionotropic receptor delta type subunit 1 which is a subunit of glutamate receptor channels that mediate the fast excitatory synaptic transmission in the central nervous system [65].
*CYP2R1* (rs12794714)	11	*CYP2R1* gene encodes the cytochrome P450 family 2 subfamily R member 1 which acts as 25-hydroxylase of vitamin D [66].
*TMEM151A* (rs61891388)	11	*TMEM151A* has been predicted to be an integral component of membrane and *CD248* enables extracellular matrix binding activity and regulates endothelial cell apoptotic process.
*AP002387.1/**ACTE1P* (rs1660839, rs12803256)	11	*ACTE1P* gene is an RNA gene. ACTE1P [67] and vitamin D [68] are both involved in adolescent idiopathic scoliosis (abnormal curvature of the spine), suggesting a possible role of ACTE1P in bone health.
*S100A11P3* (rs12798050)	11	*S100A11P3* gene encodes the S100 calcium binding protein A11 pseudogene 3. It has multiple roles in buffering calcium ion concentration, participating in energy metabolism, regulating cell proliferation and differentiation [69].
*DGAT2* (rs72997623)	11	*DGAT2* encodes the diacylglycerol O-acyltransferase 2, catalysing the synthesis of triglycerides [70]. Affects adipose tissue formation [71] with possible link to 25(OH)D storage.
*GUCY2EP/TSKU* (rs1149605)	11	*GUCY2EP* gene encodes guanylate cyclase 2E that is involved in chemosensation and *TSKU* gene encodes tsukushi, small leucine rich proteoglycan that has been predicted to act upstream/within several processes, including negative regulation of Wnt signaling pathway.
*HAL* (rs10859995)	12	*HAL* gene is upregulated during the differentiation of keratinocytes [72]. HAL deaminates L-histidine to trans-uronic acid [73], which in the stratum corneum absorbs UVB [74] and reduce the production 25(OH)D [75].
*SEC23A* (rs8018720)	14	*SEC23A* gene encodes the Sec23 homolog A, coat complex II component which plays a role in the ER-Golgi protein trafficking.
*ALDH1A2* (rs261291)	15	*ALDH1A2* gene encodes aldehyde dehydrogenase 1 family member A2 which catalyses the synthesis of retinoic acid (RA) from retinaldehyde [76].
*PDILT* (rs77924615)	16	*PDILT*/*PDIA7* gene encodes the protein disulphide isomerase like, testis expressed which catalyses protein folding and thiol-disulphide interchange reactions [77].
*SULT2A1* (rs212100)	19	*SULT2A1* gene encodes a liver- and intestine-expressed sulpho-conjugating enzyme that is responsible for the inactivation by sulphonation of 25(OH)D [78,79].
*KLK10* (rs10426)	19	*KLK10* gene encodes the kallikrein related peptidase 10 that has been shown to play a role in dermal integrity [80].
*CYP24A1* ^†^	20	*CYP24A1* gene encodes cytochrome P450 family 24 subfamily A member 1 which is an important candidate for vitamin D metabolic pathway given that it initiates the degradation of 1,25-dihydroxyvitamin D3 by hydroxylation of the side chain [81]. In addition, this enzyme also plays a role in calcium homeostasis and vitamin D endocrine system [82].
*PLA2G3* (rs2074735)	22	*PLA2G3* gene encodes the phospholipase A2 group III which functions in lipid metabolism and catalyses the calcium-dependent hydrolysis of the sn-2 acyl bond of phospholipids to release arachidonic acid and lysophospholipids [83].

* *UGT1A5*, *UGT1A6*, *UGT1A7*, *UGT1A8*, *UGT1A9*, *UGT1A10*. ^†^ rs6123359, rs17216707, rs2585442, rs2762943.

**Table 2 nutrients-14-04408-t002:** Average 25-hydroxyvitamin D level and the odds of low concentrations by quintiles in vitamin D genetic risk score in the UK Biobank.

	Vitamin D Winter (*n* = 176,577)			Vitamin D Summer (*n* = 130,855)		
	25(OH)D Mean (SD)	<25 nmol/L OR (95% CI)	<50 nmol/L OR (95% CI)	25(OH)D Mean (SD)	<25 nmol/L OR (95% CI)	<50 nmol/L OR (95% CI)
Quintile 1 (Lowest 20%)	39.16 (17.41)	Reference	Reference	52.13 (17.37)	Reference	Reference
Quintile 2	41.84 (18.47)	0.79 (0.76–0.82)	0.75 (0.73–0.78)	56.16 (18.51)	0.72 (0.66–0.79)	0.68 (0.66–0.71)
Quintile 3	43.73 (19.36)	0.68 (0.66–0.71)	0.64 (0.61–0.66)	58.50 (19.20)	0.63 (0.57–0.69)	0.56 (0.54–0.58)
Quintile 4	45.40 (20.14)	0.60 (0.58–0.62)	0.54 (0.53–0.56)	60.65 (20.05)	0.53 (0.48–0.58)	0.49 (0.47–0.51)
Quintile 5	47.51 (21.25)	0.52 (0.50–0.54)	0.47 (0.45–0.48)	64.05 (21.17)	0.42 (0.38–0.47)	0.39 (0.37–0.40)

Genetic risk score calculated using 35 variants with replicated association with 25(OH)D concentrations. Adjusted for age, sex, month in which blood sample was taken, fasting time before blood sample was taken, sample aliquots for measurement, assessment centres, SNP array, and top 40 genetic principal components. Vitamin D winter classified as November to May and vitamin D summer as June to October, based on distribution of 25(OH)D concentrations in the UK biobank [86].

## Data Availability

Original data for Table 2 is available from the UK Biobank upon application.

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
