# Peer review of "Genetic Determinants of 25-Hydroxyvitamin D Concentrations and Their Relevance to Public Health"

_nutrients, 2022, doi:10.3390/nu14204408_

Round 1

Reviewer 1 Report

Recommendations:

In figure 1, indicate what the words in yellow mean and describe the acronyms used.

The authors should make an integrating figure of the main genetic variants. Since these have their functional effect within some of the enzymes that participate in the canonical and non-canonical pathways of vitamin D, it would be very beneficial to integrate the information succinctly.

Interestingly, the genetic variants described in VDR are not associated with serum calcidiol deficiencies. Could the authors explain why? In particular, in other genetic and evolutionary studies, the FokI (SNV: rs2228570) VDR variant has been extensively studied for its functionality in its transactivation capacity, and the potential role in the hypovitaminosis D. Describe in the manuscript the arguments supporting the exclusion of VDR genetic variants in this literature review.

Author Response

In figure 1, indicate what the words in yellow mean and describe the acronyms used.

We have now revised Figure 1 and included more explicit explanations avoiding acronyms. However, all gene names continue to be presented using acronyms, as is common practice.

The authors should make an integrating figure of the main genetic variants. Since these have their functional effect within some of the enzymes that participate in the canonical and non-canonical pathways of vitamin D, it would be very beneficial to integrate the information succinctly.

As we identified the variants based on replicating findings from genomewide association studies, many of the SNPs with identified associations may not be functional, but rather reflect an effect by other (possibly unknown) correlated variants. Therefore, we fear that including rs numbers for all variants to Figure 1 may lead to confusion and also, reduce the readability of the figure. Hence, we suggest only mentioning the gene names in Figure 1 but to include an explicit reference to Table 1. This will allow the readers to access further information readily and easily as needed.

 Interestingly, the genetic variants described in VDR are not associated with serum calcidiol deficiencies. Could the authors explain why? In particular, in other genetic and evolutionary studies, the FokI (SNV: rs2228570) VDR variant has been extensively studied for its functionality in its transactivation capacity, and the potential role in the hypovitaminosis D. Describe in the manuscript the arguments supporting the exclusion of VDR genetic variants in this literature review.

This is interesting and indeed, it is a consistent observation that variants in the VDR are not associated with calcidiol concentrations. While VDR has undoubtedly a key role in mediating the metabolic actions of active hormonal vitamin D (calcitriol), this is not reflected in amount of substrate for calcitriol production (i.e. calcidiol). This likely reflects the relatively large reservoir of calcidiol (typically present in 25-100 nmol/l in the circulation) in comparison to active calcitriol, which has ~1000 times lower circulating concentrations (presented in picomoles/l in contrast to nanomoles/l). In other words, even if increased use of calcitriol would require more substrate, related differences arising from genetic variations in VDR, are not large enough to be reflected in calcidiol levels. Indeed, also variants in CYP27B1 (converting calcidiol to calcitriol) are not picked up by GWAs on calcidiol, also something that is likely to reflect the comparatively large calcidiol reservoir. This said, it is possible that these important vitamin D related variants may have an effect that is reflected on calcidiol concentrations but only when the concentrations are very low.

We have now added further points to the paper to discuss this aspect, please see lines 113-115 (document with track changes).

Reviewer 2 Report

The authors have written an interesting review article, and I appreciate their work in bringing together the literature on the genetic determinants of serum vitamin D.

MC1R

On the first two pages, the authors discuss MC1R and OCA2 as major determinants of skin color and as possible determinants of serum vitamin D. They are wrong on both counts:

·         MC1R primarily determines variation in hair color. It determines variation in skin color only in the case of alleles for red hair (which are a minority of MC1R alleles). OCA2 primarily determines variation in eye color. Variation in skin color is explained largely by alleles at SLC45A2, SLC24A5, and TYRP1 (Beleza et al. 2013).

·         The authors acknowledge that current studies have failed to find a relationship between allelic variation at MC1R and variation in serum vitamin D. They argue, however, that no relationship has been found because only Europeans were studied. The results would presumably be different if non-Europeans were also studied. The truth is that non-Europeans show very little allelic variation at MC1R (Rana et al. 1999). If the considerable variation at MC1R among Europeans has no detectable effect on serum vitamin D, why would the very limited variation at MC1R among Asians and Africans have a detectable effect?

The discussion of MC1R and OCA2 should be moved to the Discussion section of the paper. The authors could mention that genome-wide association studies have not identified either gene as a possible determinant of serum vitamin D. They could then mention that MC1R primarily affects variation in hair color and that variation in hair color is weakly associated with variation in skin color. The same is true for OCA2 and variation in eye color. This point has been made by American anthropologist Alice Brues:

“If we were to take all the human beings in the world who have dark brown eyes and black or dark brown hair, we would not only have the vast majority of the human species, but would have a group which shows virtually the complete range of human skin color, from black to almost completely depigmented.” (Brues 1975)

Personalized vitamin D supplementation

If certain individuals or groups are genetically predisposed to having low concentrations of serum vitamin D, they have probably evolved to use vitamin D more efficiently or more sparingly through higher uptake of calcium from the intestines, through increased conversion of 25(OH)D to the more active 1,25(OH)2D, through stronger binding by proteins that transport vitamin D via the bloodstream, and through greater use of alternative pathways for calcium uptake (Frost 2022). In such cases, vitamin D supplementation may have the unintended consequence of pushing vitamin D levels over the threshold of toxicity.

The last point is important. Vitamin D metabolism varies from one human population to another, as do many other aspects of physiology. This is especially the case with serum 25(OH)D: a particular level may indicate vitamin D deficiency in one population and yet be normal and healthy in another.

Quality of the writing

The writing style could be improved. There are a surprising number of spelling mistakes that could have easily been found and corrected by a spell check. Why is this so?

The following are my proposed corrections: 

Line 19 – replace “canonical” with “actual” (or simply delete)

Line 22 – delete the comma after “show”

Line 23 – replace “is” with “are”

Line 25 – replace “implementing” with “on”

Lines 26-27 – delete “have provided information that”

Line 27 – replace “impact” with “impacts”

Line 28 – replace “arising from” with “of”

Line 39 – replace “norther” with “northern”

Line 40 – replace “when having less” with “where reduction of”

Line 40 – replace “for the facilitation of the” with “to”

Lines 53-55 – replace the sentence with “For this paper, we systematically looked through the GWAS literature for genes that influence serum 25(OH)D levels.”

Line 73 – replace “seeking to identify” with “that looked for”

Line 101 – Place a period after [42] and replace “providing” with “There is thus”

Line 105 – replace the comma after “properties” with a period and replace “liver, brain and skin as” with “The liver, the brain and the skin are thus”

Line 113 – insert “meta-analyses” after “consortium”. Delete “there were” and “that”

Line 114 – replace “SUNLIGHT consortium” with “same”

Line 117 – insert “the” before “SUNLIGHT”

Line 118 – replace “relates” with “relate”. Insert “latter’s” before “body mass index”. Delete “by the SUNLIGHT”

Line 119 – replace “reducing” with “which reduces”

Line 120 – delete “of”

Line 122 – delete the comma after “function”

Line 129 – insert “meta-analyses” after “consortium”

Lines 130-131 – replace “explain the association” with “be associated”, delete “with 25(OH)D”, and replace “by affecting the available storage” with “because these tissues serve as storage sites for 25(OH)D”.

Line 132 – replace “GWAS” with “the GWASs”

Table 1, third line – replace “enzyme, which catalyze” with “enzyme that catalyzes”

Line 140 – replace “for” with “of”

Line 143 – replace “between” with “from”

Line 145 – replace “the higher the smaller the contribution of environmental factors is” with “higher with lower environmental contributions “

Line 149 – replace “smallest” with “lowest”

Line 150 – replace “compared to negilible” with “and negligible”

Line 151 – replace “in conducted in” with “of”

Lines 157-158 – replace “higer” with “higher” and “compared to” with “than in”

Lines 164-166 – replace the comma after UK Biobank” with a period and delete the remainder of the sentence.

Line 166 – replace “is” with “are”

Lines 167-168 – delete “demonstrating highly relevant impact”

Line 173 – replace “differeces” with “differences”

Line 196 – replace “combing” with “that combine”

Line 197 – replace “allels” with “alleles” and “looking” with “which look”

Line 200 – replace “somewheat” with “somewhat”

Line 211 – delete “recommendation to take”

Line 212 – replace “was” with “were”

Line 223 – replace “The” with “With the” and insert a comma after “concentrations”

Line 224 – replace “have enabled the use of” with “it has become possible to use”. Replace “in studies to examin evidence for” with “to examine”

Line 228 – replace “to” with “on”

Line 232 – replace “canonical” with “actual”

Line 234 – replace “in” with “into”

Line 238 – replace “wide-range” with “a wide range”

Line 242 – replace “One approach to” with “To”

Line 243 – replace “pleiotopry” with “pleiotropy”

Line 248 – insert “the” before “robustness”. Replace “Multivariable” with “A multivariable”

Line 262 – replace “had” with “has”

Line 264 – replace “for” with “of”

Line 267 - delete “a” and insert a comma after “fortification”

Line 269 – insert a comma after “groups”

Lines 273 – 281 – Replace “These include …” with:

Some of them are linked to the actual vitamin D metabolic pathway, and others to lipid metabolism and skin properties. In terms of methodology, they may provide MR studies with the means to measure the various determinants of serum 25(OH)D concentrations. Further research is needed to understand how such genetic information may be used to personalize vitamin D supplementation and prevent vitamin D deficiency.

References

Beleza, S., A.M. Santos, B. McEvoy, I. Alves, C. Martinho, E. Cameron, et al. (2013). The timing of pigmentation lightening in Europeans. Molecular Biology and Evolution 30(1): 24-35. https://doi.org/10.1093/molbev/mss207

Brues, A.M. (1975). Rethinking human pigmentation. American Journal of Physical Anthropology 43(3): 387-391. https://doi.org/10.1002/ajpa.1330430320

Frost P. (2022) The Problem of Vitamin D Scarcity: Cultural and Genetic Solutions by Indigenous Arctic and Tropical Peoples. Nutrients 14(19):4071. https://doi.org/10.3390/nu14194071

Rana, B.K., D. Hewett-Emmett, L. Jin, B.H.J. Chang, N. Sambuughin, M. Lin, et al. (1999). High polymorphism at the human melanocortin 1 receptor locus. Genetics 151(4): 1547-1557.

Author Response

The authors have written an interesting review article, and I appreciate their work in bringing together the literature on the genetic determinants of serum vitamin D.

MC1R

On the first two pages, the authors discuss MC1R and OCA2 as major determinants of skin color and as possible determinants of serum vitamin D. They are wrong on both counts:

  • MC1Rprimarily determines variation in hair color. It determines variation in skin color only in the case of alleles for red hair (which are a minority of MC1R alleles). OCA2 primarily determines variation in eye color.

Variation in skin color is explained largely by alleles at SLC45A2SLC24A5, and TYRP1 (Beleza et al. 2013).

          The authors acknowledge that current studies have failed to find a relationship between allelic variation at MC1R and variation in serum vitamin D. They argue, however, that no relationship has been found because only Europeans were studied. The results would presumably be different if non-Europeans were also studied. The truth is that non-Europeans show very little allelic variation at MC1R (Rana et al. 1999). If the considerable variation at MC1R among Europeans has no detectable effect on serum vitamin D, why would the very limited variation at MC1R among Asians and Africans have a detectable effect?

The discussion of MC1R and OCA2 should be moved to the Discussion section of the paper. The authors could mention that genome-wide association studies have not identified either gene as a possible determinant of serum vitamin D. They could then mention that MC1R primarily affects variation in hair color and that variation in hair color is weakly associated with variation in skin color. The same is true for OCA2 and variation in eye color. This point has been made by American anthropologist Alice Brues:

“If we were to take all the human beings in the world who have dark brown eyes and black or dark brown hair, we would not only have the vast majority of the human species, but would have a group which shows virtually the complete range of human skin color, from black to almost completely depigmented.” (Brues 1975)

As a general response, we are somewhat surprised by the above comment suggesting we were ‘wrong’, since the reviewer themselves appears to acknowledge that both OCA2 and MC1R do affect skin colour (for both variants this is also confirmed by scientific literature), and the statement which we included was simply “Genes such as OCA2, MC1R or SLC24A4 which affect human skin colour have great interest as candidate variants for serum 25(OH)D.”  The “great interest” is clearly a perspective which is more subjective, and we appreciate that what we feel is ‘of great interest’ may not have been such for this reviewer (but yet, this does not make us ‘wrong’). However, respecting the reviewers expertise and clearly strong sentiment, we have now deleted references to OCA2 and MC1R from this sentence, which should remove any controversy around the area.

RE: MC1R, if we accept that this variant affects skin colour, and that certain variants are seen in darkly pigmented population groups  (with little variation between individuals within this group), and that different variants are carried by individuals with lighter skins; by default analyses which would combine groups with different skin colours would be expected to detect a the related genetic difference in skin colour. Consequently, if this difference in skin pigmentation is large enough, then it should by default associate with vitamin D synthesis in the skin, leading to differences in serum 25(OH)D levels. The reason why no difference has been seen in analyses constrained to white Europeans may be because in statistical analyses, it is always problematic to detect differences if we are investigating associations within groups which are too homogenous.

However, we have now much simplified the related notes in the introduction, where this information was used merely to ‘set the scene’. We prefer not to include further discussion in the paper to avoid retractions from the focus. For related changes please see page 2, document with track changes).

Personalized vitamin D supplementation

If certain individuals or groups are genetically predisposed to having low concentrations of serum vitamin D, they have probably evolved to use vitamin D more efficiently or more sparingly through higher uptake of calcium from the intestines, through increased conversion of 25(OH)D to the more active 1,25(OH)2D, through stronger binding by proteins that transport vitamin D via the bloodstream, and through greater use of alternative pathways for calcium uptake (Frost 2022). In such cases, vitamin D supplementation may have the unintended consequence of pushing vitamin D levels over the threshold of toxicity.

 The last point is important. Vitamin D metabolism varies from one human population to another, as do many other aspects of physiology. This is especially the case with serum 25(OH)D: a particular level may indicate vitamin D deficiency in one population and yet be normal and healthy in another.

Thank you for this interesting point. We have now added a related note in the context of personalised approaches, including a reference to Frost 2022 paper (lines 242-246, document with track changes).

Quality of the writing

The writing style could be improved. There are a surprising number of spelling mistakes that could have easily been found and corrected by a spell check. Why is this so?

 Thank you for this point, which we clearly should have picked up ourselves. The spell check function was turned off by using the template for Nutrients, which is not something we expected. Our apologies.

The following are my proposed corrections: 

 Thank you for your scrutiny, which has clearly improved the quality of presentation. We have actioned most of your comments as suggested, with few exceptions as noted below.  

Line 19 – replace “canonical” with “actual” (or simply delete)

done

Line 22 – delete the comma after “show”

done

Line 23 – replace “is” with “are”

done

Line 25 – replace “implementing” with “on”

done

Lines 26-27 – delete “have provided information that”

done

Line 27 – replace “impact” with “impacts”

done

Line 28 – replace “arising from” with “of”

done

Line 39 – replace “norther” with “northern”

done

Line 40 – replace “when having less” with “where reduction of”

done

Line 40 – replace “for the facilitation of the” with “to”

done

Lines 53-55 – replace the sentence with “For this paper, we systematically looked through the GWAS literature for genes that influence serum 25(OH)D levels.”

done

Line 73 – replace “seeking to identify” with “that looked for”

done

Line 101 – Place a period after [42] and replace “providing” with “There is thus”

done

Line 105 – replace the comma after “properties” with a period and replace “liver, brain and skin as” with “The liver, the brain and the skin are thus”

We don’t fully agree with this change, and have now revised the sentence to read “ …and pathways related to skin properties. The  liver, the  brain and the skin were identified as the top three…”

Line 113 – insert “meta-analyses” after “consortium”. Delete “there were” and “that”

We have inserted “ meta-analyses” but otherwise kept the sentence in original form for explicit clarity. L

ine 114 – replace “SUNLIGHT consortium” with “same”

Not actioned, but kept in original format for clarity.

Line 117 – insert “the” before “SUNLIGHT”

Done

Line 118 – replace “relates” with “relate”. Insert “latter’s” before “body mass index”. Delete “by the SUNLIGHT”

We have actioned the first part, but kept the latter part relating to SUNLIGHT consortium for clarity.  

Line 119 – replace “reducing” with “which reduces”

done

Line 120 – delete “of”

done

Line 122 – delete the comma after “function”

done

Line 129 – insert “meta-analyses” after “consortium

done

Lines 130-131 – replace “explain the association” with “be associated”, delete “with 25(OH)D”, and replace “by affecting the available storage” with “because these tissues serve as storage sites for 25(OH)D”.

done

Line 132 – replace “GWAS” with “the GWASs”

We feel this would not be correct, given in this context we are referring to variants that were identified in one GWAS but replicated in an another, not to variants that were identified by multiple independent GWASs.

Table 1, third line – replace “enzyme, which catalyze” with “enzyme that catalyzes”

done

Line 140 – replace “for” with “of”

done

Line 143 – replace “between” with “from”

done

Line 145 – replace “the higher the smaller the contribution of environmental factors is” with “higher with lower environmental contributions “

done

Line 149 – replace “smallest” with “lowest”

done

Line 150 – replace “compared to negilible” with “and negligible”

done

Line 151 – replace “in conducted in” with “of”

done

Lines 157-158 – replace “higer” with “higher” and “compared to” with “than in”

done

Lines 164-166 – replace the comma after UK Biobank” with a period and delete the remainder of the sentence.

done

Line 166 – replace “is” with “are”

done

Lines 167-168 – delete “demonstrating highly relevant impact”

done

Line 173 – replace “differeces” with “differences”

done

Line 196 – replace “combing” with “that combine”

done

Line 197 – replace “allels” with “alleles” and “looking” with “which look”

done

Line 200 – replace “somewheat” with “somewhat”

done

Line 211 – delete “recommendation to take”

It would not be appropriate to delete this, given we are not referring to a RCT. The only ‘intervention’ was a recommendation rather than any form of supplementation. Kept as is.

Line 212 – replace “was” with “were”

Not changed, ”was” correct in this context.

Line 223 – replace “The” with “With the” and insert a comma after “concentrations”

done

Line 224 – replace “have enabled the use of” with “it has become possible to use”. Replace “in studies to examin evidence for” with “to examine”

done

Line 228 – replace “to” with “on”

done

Line 232 – replace “canonical” with “actual”

done

Line 234 – replace “in” with “into”

done

Line 238 – replace “wide-range” with “a wide range”

done

Line 242 – replace “One approach to” with “To”

Not changed, sentence correct as is.

Line 243 – replace “pleiotopry” with “pleiotropy”

done

Line 248 – insert “the” before “robustness”. Replace “Multivariable” with “A multivariable”

done

Line 262 – replace “had” with “has”

Correct as is; “had” refers to linear MR studies, which had already provided evidence before the non-linear approaches were implemented.

Line 264 – replace “for” with “of”

done

Line 267 - delete “a” and insert a comma after “fortification”

done

Line 269 – insert a comma after “groups”

done

Lines 273 – 281 – Replace “These include …” with:

Some of them are linked to the actual vitamin D metabolic pathway, and others to lipid metabolism and skin properties. In terms of methodology, they may provide MR studies with the means to measure the various determinants of serum 25(OH)D concentrations. Further research is needed to understand how such genetic information may be used to personalize vitamin D supplementation and prevent vitamin D deficiency.

 done